# A contingent valuation experiment about future particle accelerators at CERN

**Massimo Florio** [1]*, **Francesco Giffoni** [1,2]

1 Department of Economics, Management, and Quantitative Methods, University of Milan, Milan, Italy,
2 CSIL–Centre for Industrial Studies, Corso Monforte, Milan, Italy

* massimo.florio@unimi.it

## Abstract

Investment in basic science is mainly supported by government funding, but little is known about citizens' willingness to pay for large-scale projects. A survey to a representative sample of French taxpayers, designed as a contingent valuation experiment about a future particle accelerator for CERN, reveals that citizens' willingness to pay is correlated with education, income, age, and–crucially–previous awareness, attitudes and interest in science. A (slim) majority of the participants would accept paying more in taxes for CERN. The estimated willingness to pay is higher than the current implicit per capita tax burden of French citizens. The experimental setting is novel and replicable for empirically assessing social attitudes towards science for other research infrastructures and countries.

**Data Availability Statement:** All relevant data are within the paper and its Supporting Information files. Data are also available from http://www.eiburs.unimi.it/.

**Funding:** This work was carried out in the framework of the FCC study collaboration agreement [grant number KE3044/ATS] between

## Introduction

CERN is the European Organization for Nuclear Research and it is based in Geneva, Switzerland. Established in 1954, today the laboratory includes 23 Member States, which financially support its research activity in particle physics by annual contributions. CERN's main mission is to perform frontier research in fundamental physics to analyse what the universe is made of and how it works. It currently operates several accelerators, including, notably, the Large Hadron Collider (LHC), the world's largest and most powerful particle accelerator. In 2019, CERN unveiled the conceptual design report for a Future Circular Collider to be based in a 100 km underground tunnel at the border between France and Switzerland. The construction cost of this post-Large Hadron Collider accelerator (the stand-alone FCChh version) is estimated at EUR 24 bn [1], which is about six times greater than that of the LHC. The Future Circular Collider is just one example of the cost of contemporary Big Science. In 2018, China developed a conceptual design of a similar project [2]. ITER, the fusion test reactor in France, supported by a collaboration of 35 nations, has an estimated construction cost of EUR 19 bn [3]. Public funding of basic science projects raises many questions [4,5,6], not least whether taxpayers are actually willing to pay for such investments. For example, every US citizen aged above 18 years contributes on average more than USD 90 per year to space exploration and other NASA activities [7]. In 2019, the annual contributions of Member States to CERN amounted to EUR 0.9 bn [8], meaning that every adult citizen in CERN Member States contributed EUR 2.7 to CERN each year.

the European Organization for Nuclear Research (CERN) and the University of Milan. A matching grant to the Department of Economics, Management, and Quantitative Methods was offered by the University of Milan research funds. The funders had no role in study design, data collection and analysis, decision to publish, or preparation of the manuscript.

**Competing interests:** The authors have declared that no competing interests exist.

While Big Science investments in basic research are likely to advance human knowledge and address societal challenges, the growing costs of such large-scale investments are creating controversy in the science policy arena given increasing governments budget constraints and the fact that citizens, as taxpayers, are the ultimate funders of Big Science investments. This article explores preferences of citizens about funding science, a little-known field of study [9, 10]. Basic research and its discoveries, such as the Higgs boson, the gravitational waves or having taken the first ever image of a black hole, usually have no direct and tangible impact on utility for the average taxpayer. So what drives citizens to support Big Science projects financially? Does a non-use value for science as a public good exist [11], as opposed to use-values generated by more tangible scientific applications, i.e. in medical applied research? Our research, therefore, addresses the question of the extent to which the pursuit of apparently "useless" discoveries for the public outside the scientific community may generate some form of satisfaction or curiosity, which in turn manifests in willingness-to-pay (WTP) for fundamental science.

To reach our goal, this study makes use of a contingent valuation (CV) survey, a stated preferences technique, which has been extensively used as an empirical research method to ask and quantify individual preferences for (public) goods, including environmental goods [12], cultural goods [13], and broadly defined public goods [14]. The CV approach was discussed in depth after the EXXON Valdez accident occurred in 1989 and guidelines were provided by the NOAA's Panel in 1993 [15]. Since then, a large number of CV studies have followed and, while there still are controversies surrounding the use of stated preferences in economics, a consensus has emerged about how to design a CV experiment [16]. We extend the CV approach to scientific projects by designing and implementing a CV survey about CERN future investments. The survey was administered to a representative sample of 1,005 French taxpayers and points to a core research question: Is citizens' willingness to pay for fundamental science compatible with their actual tax burden?

The rest of paper is structured as follows. We begin with the background of the study and describe its design phase. We then present the results and discuss their implications. The Supporting Information provides extra documentation and details on pilot experiments, pre-tests and the survey development. It also contains an extensive description of the socio-demographic profile and cultural traits of the participants in the experiment and supplementary analyses supporting the evidence in the main text.

## Materials and methods

### Background

The CV experiment on social attitudes about CERN future accelerators was designed as a survey of a representative sample of French citizens between 18 and 74 years old and implemented after more than three years of design activities, pilot experiments, and pre-tests in four countries (S1 File).

The methodological choice to use CV, wherein respondents are asked whether they would pay for a suggested change at a given cost, relies on the fact that this technique has been traditionally applied to measure the WTP for changes in public goods or other goods creating externalities, including ecosystems, environmental programs, human health services, and recently for the valuation of cultural goods such as museums, media broadcasting, cultural heritage, and arts in general [12, 13, 17, 18]. This spectrum of applications reflects a sustained extension in the use and importance of results from CV studies to inform decision making and detect non-use values [19]. In a similar vein, this research uses CV and its flexibility to construct

scenarios for the evaluation of (changes in) large-scale projects in basic science with relevant spill-overs, such as particle accelerators [5, 20], for which stated preference data are still scarce.

The idea of using surveys to evaluate non-market goods can be traced back to [21] and first applied by [22] to elicit the value of outdoor recreation. The closed-ended single-bounded dichotomous choice (SBDC-) CV format was, on the other hand, introduced by [23] (see below). Despite this long tradition and wide range of applications, the validity of CV studies in capturing real preferences of people is still debated [16]. Concerns mainly focus on survey development and design as well as data analysis. Survey development and design cover relevant issues such as ethics and protection of human subjects, hypothetical bias (where respondents may not give true or reasonable answers with respect to contingent scenarios), policy consequentiality (i.e. making respondents aware that the results of the survey will influence actual policy), the amount of bids to be asked, the payment vehicle, and incentive-compatible WTP elicitation procedures, for instance open-ended elicitation versus closed-ended valuation questions such as SBDC-CV or double-bounded dichotomous choice (DBDC-) CV question format. As far as data analysis is concerned, several econometric modelling alternatives may be available, both with parametric and non-parametric models. Guidelines on how to address major decisions in the design and implementation of a stated preferences study, including CV, have been proposed in earlier literature. The present research was grounded, as far as possible, in the most recent guidelines for CV [16], which is more comprehensive than the seminal consensus report provided in 1993 by the National Oceanic and Atmospheric Administration (NOAA) Blue Ribbon Panel on CV [15]. In the following sections, the questionnaire design and choices made associated with the above themes are explained in detail.

## Design phase

The design phase of the questionnaire submitted to the sample of French citizens started in August 2017 and took advantage of the experience accumulated during previous pilot experiments. The design phase was conceived as a two-step process.

In the first step, meetings between CERN staff, the team of the University of Milan, and external CV experts focused on the objectives and content of the questionnaire. A crucial survey design issue was how to describe CERN, its research activity, and the possible ways particle physics research at CERN can impact on society. This step was followed by desk research to further substantiate the information set addressed to respondents.

Two alternative scenarios were introduced by a preamble informing respondents about the progress in fundamental physics that particle accelerators have permitted so far, including the discovery of the Higgs boson. We looked for scientifically accurate, policy relevant, and understandable contingent scenarios. It was then decided to give respondents two realistic investment scenarios with a well-defined asset to be valued (i.e. continuing research with new projects in particle accelerators as described by the investment "Scenario A" versus the non-investment "Scenario B") and a plausible time horizon within which the investments are expected to be operational. The investment "Scenario A" (which was the change being evaluated) was therefore opposed to a non-investment "Scenario B" (the status quo), and respondents were asked to pay for the investment necessary for "Scenario A" through a tax increase to make explicit the fact that only government funds enable CERN to continue its research activity. How much on average a French citizen currently implicitly pays to CERN per year through taxes was not revealed to avoid anchoring effects. The mechanism of the CV experiment is shown in Fig 1.

The dichotomous choice CV, both in the single and the double-bounded formulation, is the most common technique among practitioners of contingent valuation. The single-bounded

PARTICLE ACCELERATOR RESEARCH, INCLUDING THE LARGE HADRON COLLIDER (LHC) AT CERN, HAS ESTABLISHED A THEORETICAL REPRESENTATION OF THE UNIVERSE. HOWEVER, THE RESEARCH HIGHLIGHTS PHENOMENA THAT CANNOT BE EXPLAINED BY THIS THEORY.

CERN MEMBER STATES, INCLUDING FRANCE, ARE FINANCING THIS RESEARCH. HERE ARE TWO POSSIBLE SCENARIOS FOR THE FUTURE OF SUCH A RESEARCH

### SCENARIO A

*CERN Member States decide to invest in a new particle accelerator in the next decade. It will make discoveries on phenomena that cannot be explained today. This new accelerator will be operated for at least twenty-five years.*

### SCENARIO B

*CERN Member States decide not to invest in a new particle accelerator. The research activity with the existing accelerator, the LHC, will gradually decrease over the next twenty years. The possibility of finding answers on unexplained phenomena will remain limited.*

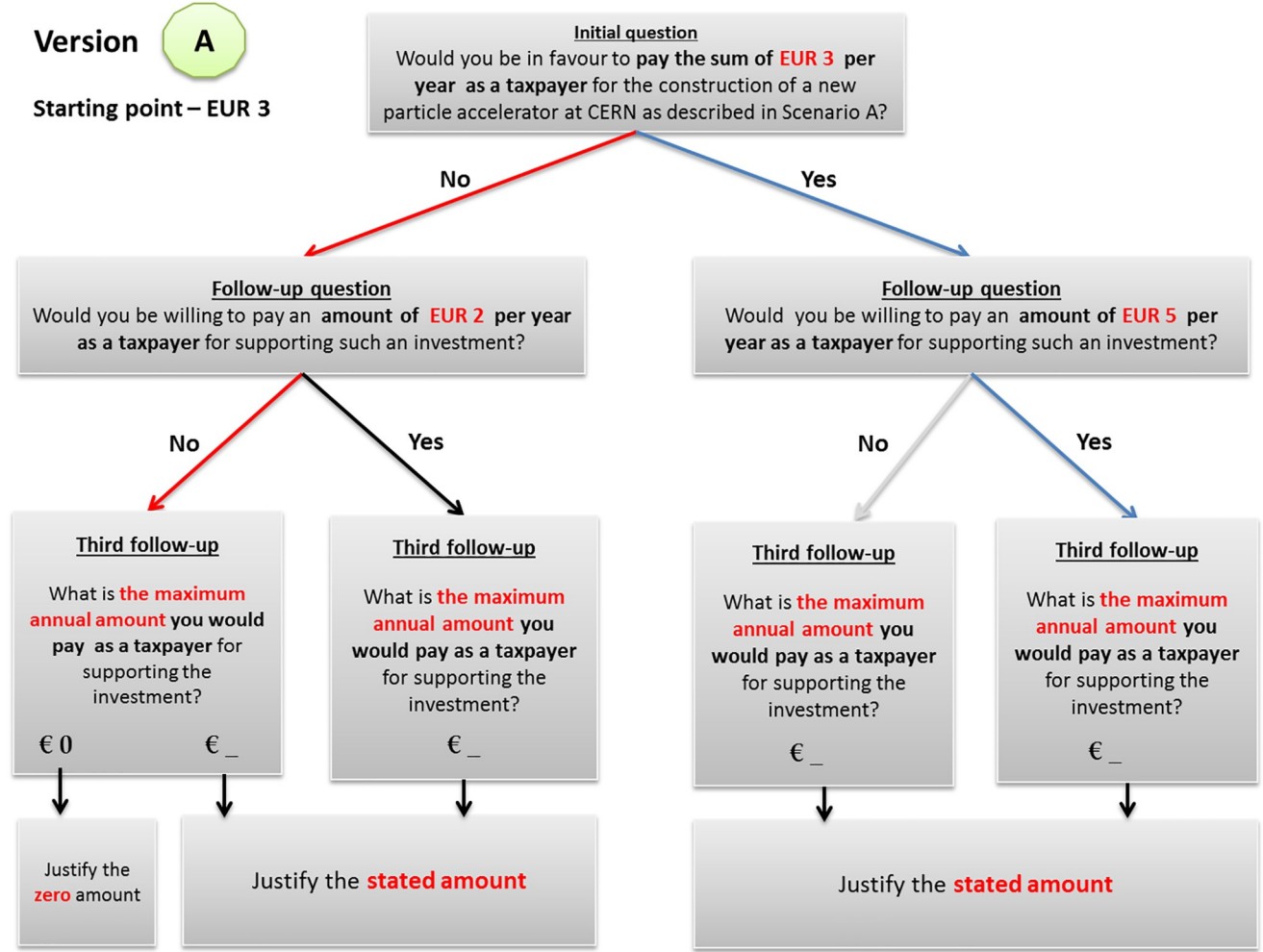

**Fig 1. The preamble, scenarios, and the elicitation procedure introduced to French citizens to elicit their WTP for investments in particle physics research carried out at CERN.** "Scenario A" is the investment scenario and "Scenario B" is the non-investment scenario. The figure displays as an example version A of the questionnaire with the following triple combination of bids: EUR 3 (the initial one), EUR 2 (the lower one), and EUR 5 (the higher one). This is the English translation; original submission was in French.

dichotomous choice (SBDC-CV), introduced in 1979 [23], was later recommended by the NOAA's Panel on CV [15] as the preferred way of designing evaluation questions. According to this framework, respondents are only required to answer "yes" or "no" when asked to pay a given amount (bid) for the non-market good and it is claimed that this procedure better mirrors the market functioning and is incentive compatible. Other available elicitation procedures (double-bounded and other multiple-bounded formats) are likely to violate incentive compatibility, but it is well known that they yield more efficient WTP estimators than the single-bounded estimator [24]. Actually, according to [16], multiple evaluation questions are likely to not affect incentive compatibility of the initial question if respondents are unaware that they will be asked subsequent valuation questions. Hence, the elicitation format employed in this study is the double-bounded dichotomous choice (DBDC-CV). In this framing, the bid level offered in the follow-up questions is higher than the offered bid in the initial payment if the answer to the initial payment question is "yes"; in contrast, if the answer is "no" a lower bid is offered. The DBDC-CV format yields four answers paths: "yes-yes" (both answers are "yes"); "yes-no" (a "yes" followed by a "no"); "no-yes" (a "no" followed by a "yes") and "no-no" (both answers are "no"). Moreover, in order to further detect lower or upper bounds of individual WTP, a third follow-up question on the maximum WTP is asked according to the scheme shown in Fig 1, which is not new in CV literature [25]. Whatever the path followed by respondents, closed-ended questions were asked to justify the declared WTP and identify protest answers.

The ultimate goal in this first round was to develop a valid survey instrument providing sufficient and unbiased information to make an informed decision, but without providing excess information that respondents think they do not need. The resulting questionnaire and sampling procedure are described below.

In the second step, the polling company Eumetra MR S.r.l. (https://www.eumetramr.com/en/eumetra-mr) was selected after a call for tender with other qualified invited participants. Selection criteria included proven competences with references and at least 10 years of experience in the following topics: market and opinion surveys; behaviour and attitudes surveys of citizens, and consumers; collecting, processing, and providing quantitative and qualitative research data; construction of samples that are nationally representative and which permit the analysis of specific target groups; having access to a panel of respondents in France. Once selected, the polling company, CERN (International Relations Sector) and several scientists involved in different research programs were involved in meeting the requirements of clarity and understandability of the questionnaire for the public. The objective was to ensure, as far as possible, that respondents from all educational levels, ages, and different life experiences would be able to comprehend the language, concepts, and questions used in the survey. To this end, after extensive pre-testing, a two-page document with photos and a two-minute movie on what CERN is and what its particle physics research consists of were designed (see below). This information set was part of the experiment and submitted to respondents before asking them the WTP questions. Moreover, open-ended and diagnostic questions at various points in the questionnaire were used to double check the understanding of the questions.

Since the outset, subsequent drafts of the questionnaire were circulated among experts at University of Milan, CERN, and market research experts to fine-tune the survey design and to conduct trial interviews. The questionnaire was repeatedly revised to improve clarity, plausibility and meaningfulness so as to enhance credibility before fifteen face-to-face pre-tests were

carried out in different locations in France. After each interview data were analysed, and the questionnaire was further revised on the basis of the analysis and interviewer debriefings so as to allow the final version of the questionnaire in French to be drafted.

## The questionnaire

The final version of the questionnaire (S2 File) was structured in four sections, addressing, respectively:

Section A "*Your interests*". It investigated interviewees' general interests and their opinion about the importance of scientific research. Questions about the knowledge and personal perception of CERN were asked as well;

Section B "*Your knowledge of CERN*" focused on CERN and its research activity. Interviewees were provided with a two–page description of CERN and a movie (hereafter referred to as information material, S3 File) so as to give them common information about particle physics research at CERN and its socioeconomic impact, the functioning of the accelerators system, and the answers that scientists expect from this research facility. Specifically, the two-page document firstly explained CERN, where it is based, its governance, and its mission. Afterwards, the CERN particle accelerators system and how it works was introduced. Finally, respondents were made aware that several technologies developed at CERN have found applications in other areas, including materials science, biology, medicine, archaeology, chemistry, and national security. Descriptions were supported by fourteen photos. The movie explained these concepts further with the help of 3D-images and footage of real events at CERN. Respondents' subjective judgements about the aspects of CERN they appreciate (or do not) and general comprehension of its research activity were checked with both open- and close-ended questions;

Section C "*Your support at CERN*" contained questions aiming at eliciting respondents' WTP for particle physics research at CERN;

Section D "*Your profile*" included questions on the respondents' demographic and socio-economic characteristics.

## Ethics

This study complies with the regulations stipulated by the Ethics Committee of University of Milan (Dean Decree of July, 19 2011) and it was approved on October, 31th 2017 (Decision 36/2017 of the Ethics Committee of the University of Milan). The study is also in compliance with the International Code on Market and Social Research regarding the respect of privacy (Code ICC/ESOMAR), with the French law on information technology, data files and civil liberties (Law number 78–17 of January 6, 1978) and CERN's Operational Circular n˚ 11 (OC11) on the processing of personal data. Respondents participated voluntarily by signing an informed consent form (S4 File) that notified them, among other things, of the electronic storage of the answers with access granted to researchers at University of Milan and CERN, and the use of anonymized data only for scientific purposes.

## Truth telling

A key challenge in stated preferences surveys is that of getting people to tell the truth. The "hypothetical bias" is the difference between values people say that they are willing to pay through hypothetical methods (e.g. in the case of the contingent investment "Scenario A") and what an individual might actually pay for the provision of the good, as revealed in non-hypothetical settings. One possible reason for poor truth elicitation is that respondents may not take the valuation experiment seriously because it is a hypothetical scenario. The guidance for SP studies [16] recommends encouraging truthful responses and suggests different

approaches. Ex-ante hypothetical bias mitigation measures such as solemn oaths, honesty priming, social responsibility techniques, and cheap talks (where explicit warnings about the problem of hypothetical bias is provided to respondents), have been developed in social psychology and try to build on the reasons why hypothetical bias appears to discipline revelation before it takes place [26, 27, 28]. In order to mitigate the risk of hypothetical bias, this study employed oath and social responsibility methods [26, 27]. Specifically, the following sentences were added to the informed consent:

> "*You are among the people selected to constitute a sample of the French population and you are invited to express yourself on this subject. Your participation is greatly appreciated, and your sincere opinion can help make decisions about strengthening research activities at CERN*".

Yet, respondents were invited to give their consensus, among other things, on the following sentence:

> "*I understand the relevance of this survey for CERN and for its investments in future research activities. I therefore undertake to provide honest and sincere answers*".

### Policy consequentiality

Policy consequentiality is strictly connected to the truth telling issue and it is commonly used for improving the realism of the survey [29] by conveying to respondents the idea that their own choice might actually impact the CERN investment program. To meet this point and beyond the sentences above, the informed consent listed a brief explanation of the purpose and the contents of the interview. Additionally, the questionnaire clarified the context of the investment decision by providing general background information on CERN, how it is funded, and how the results of the survey will be disseminated. Respondents were informed that CERN is an intergovernmental organization funded by its Member States through taxation and that CERN was conducting the CV study to advise its Member States' policy making and funding agencies. Consequentiality is also affected by the elicitation method; i.e. by the plausibility of bids and details about how the good to be evaluated would be provided.

### Bids and the payment vehicle

The guidance for SP studies does not discuss in detail how to select bid levels in CV questions and suggests making use of information from prior empirical research, pre-testing, and consider what the literature suggests [16]. Bids submitted to respondents were the results of the pilot experiments, pre-tests and consultation with polling experts. In parallel with these activities, we did a meta-analysis of 65 contemporary studies in different sectors related to the provision of public goods (environment, cultural goods, science, and technology) to have some benchmark of the bid asked and resulting mean and median WTPs. The selected studies asked for the individual WTP in the form of a yearly payment. The meta-analysis indicated a mean WTP of EUR 23.53 per person per year (median EUR 16.09) with a standard deviation of 24.73. The minimum value was EUR 1.50 and the maximum value recorded was EUR 125.23.

On the basis of these activities, five versions of the questionnaire (A, B, C, D, E) were produced, which only varied in the amount of the pre-printed initial (and subsequent) bids asked. The initial bid and the respective lower and upper bids are reported in Fig 2.

As far as the payment vehicle is concerned, it is required to be realistic, credible, familiar, and binding for all respondents, i.e. non-voluntary [16]. This would enhance incentive

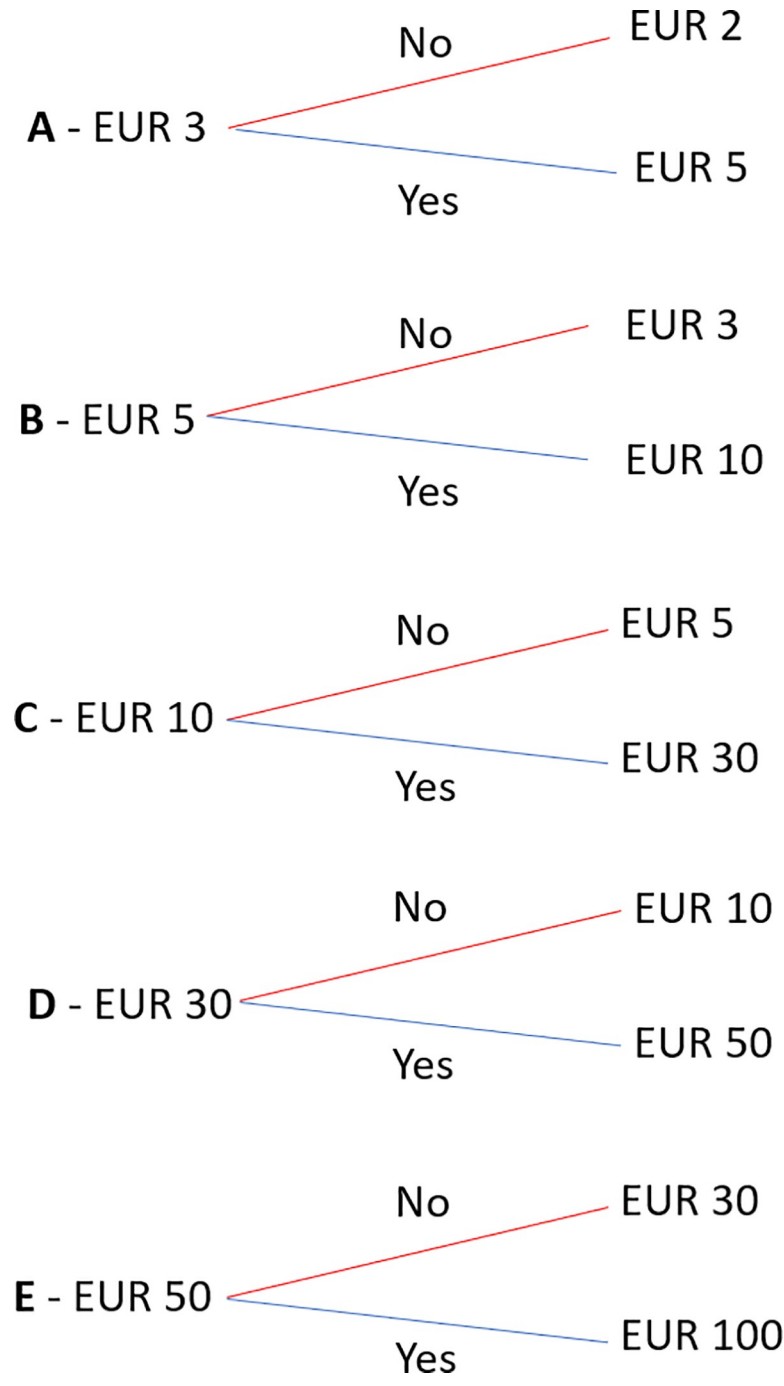

**Fig 2. Bidding scheme.**

compatibility and reduce free riding. The payment vehicle used in this study was taxation, which has the advantage of meeting all the requirements above and it is consistent with the fact that CERN is funded through Member States' contributions, which ultimately come from taxes. The disadvantage is that generally people are particularly averse to tax increases, and hence "no" answers are encouraged by selecting this payment vehicle. It is likely that different payment vehicles (e.g. donations) would have had different distributional effects on the WTP

for research investments at CERN, but we were interested in taxation as this is the appropriate mechanism in our context, and we prefer in any case to be conservative in the estimation.

The majority of CERN's budget is spent on the construction and operation of its scientific installations, which are available for use by all the participating countries. The allocation of the budget is approved on an annual basis by the CERN Council. Accordingly, the payment was asked on an annual and individual basis.

## Estimation procedure

This section addresses the utility theoretic structure assumed for the WTP estimation within the particular context of this study. Different WTP distribution assumptions were tested by using both parametric and non-parametric models [16, 30], either including covariates or not. The reason is that the mean WTP is sensitive to the estimation procedure and hypotheses about the random component of the individual WTP. Starting with the random utility difference model developed by [31] and first applied by [32] for CV, the subsequent development for DBDC-CV surveys in [24] is discussed. Yet, the distribution of the data to be analysed motivates the discussion of two additional estimation procedures: the spike model [33] and a non-parametric model based on the maximum WTP stated by respondents.

**The double-bounded dichotomous choice (DBDC-CV) model.** There are N independent respondents indexed by i = 1,...,N, whose utility function consists of some observable components (e.g. sex, income, education, age, scientific interests and so on) and components that cannot be observed by the analyst, and therefore the respondent's utility function can be written as:

$$U_i(X_{1i}, X_{2i}, S_0) + u_i \tag{1}$$

where $X_{1i}$ denotes the income, $X_{2i}$ is a vector of socio-economic characteristics, $S_0$ is the current state of the asset under evaluation (i.e. the existing LHC accelerator at CERN as described by the non-investment "Scenario B") and $u_i$ is an identically and independently distributed (i.i.d.) random error term with zero expected value. When a change in $S_0$ (from $S_0$ to $S_1$) is asked at the specific bid ($t_i$), the respondent will accept the offer if:

$$U_i(X_{1i} - t_i, X_{2i}, S_1) + u_{i1} \geq U_i(X_{1i}, X_{2i}, S_0) + u_{i0} \tag{2}$$

where $S_1$ indicates the investment "Scenario A", and $u_{i0}$ and $u_{i1}$ are i.i.d. While from the viewpoint of the respondent the choice is the result of an utility maximizing process, for the researcher the response is a random variable with some cumulative distribution function (c.d.f.) denoted by $G(t_i; \theta)$, where $\theta$ is a vector of parameters of the distribution to be estimated on the basis of the responses to the CV survey. Specifically, the probability that a respondent will accept paying the bid $t_i$ and recording a "yes" answer is given by (the conditioning on $X_{1i}$, $X_{2i}$, and $S_j$, $j = 0,1$ is omitted for simplicity):

$$Prob\ (\text{yes}) = Prob\ (t_i \leq WTP_i) = 1 - G(t_i;\ \theta) \tag{3}$$

In the DBDC-CV setting, each respondent is asked for two subsequent bids. Let $t_i$ denote the generic bid asked of respondents $i$ and $t_i^0$ the initial bid. If the respondent answers "yes" to the initial bid, the upper bound follow-up bid $t_i^u > t_i^0$ is asked; otherwise she receives the lower bound follow-up bid $t_i^l < t_i^0$. This format leads to four possible answer paths: "yes-yes", "yes-no", "no-yes", "no-no". Let $I_i^{YY}, I_i^{YN}, I_i^{NY}, I_i^{NN}$ be indicator functions such that:

$$I_i^{YY} = \boldsymbol{I}\ (ith\ respondents'path\ is\ "yes - yes") \tag{4}$$

$$I_i^{YN} = \boldsymbol{I}\ (ith\ respondents'path\ is\ "yes - no")$$

$$I_i^{NY} = \mathbf{I} \ (ith \ respondents' path \ is \ "no - yes")$$

$$I_i^{NN} = \mathbf{I} \ (ith \ respondents' path \ is \ "no - no")$$

where $\mathbf{I}(.)$ takes on the value of one if its argument is true and zero otherwise. The maximum likelihood estimator $\hat{\theta}$ is obtained by maximizing the following log-likelihood function:

$$ln \, L(\theta) = \sum_{i=1}^{N} \{I_i^{YY} \, ln[1 - G(t_i^u; \ \theta)] + I_i^{YN} \, ln[G(t_i^u; \ \theta) - G(t_i^0; \ \theta)] + I_i^{NY} \, ln[G(t_i^0; \ \theta) - G(t_i^l; \ \theta)] + I_i^{NN} \, ln G(t_i^l; \ \theta)\} \quad (5)$$

In Eq 5, the terms in brackets associated with the indicator functions denote the probability of observing the respective event. For instance, the term $[1 - G(t_i^u; \ \theta)]$ associated with $I_i^{YY}$ indicates the probability of observing the answer path "yes-yes", similarly the term $[G(t_i^u; \ \theta) - G(t_i^0; \ \theta)]$ denotes the probability of observing a "yes-no" answer and so on. Since the indicator functions take the value of one or zero according to the answers of each individual, it is as to say that a given individual contributes to the logarithm of the likelihood function in only one of its four parts.

The way the empirical mean WTP is obtained depends upon a set of assumptions on the functional from of individual utility in Eq 1 and the statistical distribution of $u_i$ in Eq 1 or equivalently of $G(t_i; \theta)$ in Eq 3. When the utility function is composed of a linear deterministic component and a stochastic element such as:

$$U_i(X_i, S_j) = X_i\beta + u_{ij}; J = 0, 1 \quad (6)$$

where $X_i$ includes income, other socioeconomic characteristics and the constant term which is a function of $S$, then the WTP is given by:

$$WTP_i(X_i, u_{ij}) = X_i\beta + \Delta u_{ij} \quad (7)$$

with $\Delta u_{ij} = u_{i1} - u_{i0}$ In this setting, the expected WTP equals the median, and it can be empirically obtained as [24, 31]:

$$Mean \ (WTP) = -\frac{\bar{X}\hat{\beta}_1}{\hat{\beta}_{bid}} \quad (8)$$

where $\bar{X}$ is a vector containing the sample means of socioeconomic characteristics plugged into the model and also includes 1 for the constant term; $\hat{\beta}_1$ is a vector of the estimated coefficients and $\hat{\beta}_{bid}$ is the coefficient on the bid $t_i$ (in constant-only models, $\bar{X} = 1$ and $\hat{\beta}_1$ is the coefficient on the constant term).

Before estimating the model, the statistical distribution of $u_i$, or $G(t_i; \theta)$, has to be assumed as well [30]. Assuming symmetric distributions for $u_i$ such as normal, in that case $G(t_i; \theta)$ is the standard normal c.d.f. $\Phi \ (t_i; \theta)$, or logistic, in that case the logistic c.d.f. is $G(t_i; \theta) = (1+\exp(-X_i\beta))^{-1}$, the range of WTP is the real line and nothing guarantees that the estimated mean is positive. Therefore, for particular values of $\bar{X}$, the mean WTP may be negative. Very often, however, negative estimates of WTP do not represent true preferences of people [33] (i.e. if the person dislikes the public good or she is indifferent, she may rather ignore that good), but they arise because of the wrong assumed distribution for the c.d.f. As a result, alternative estimation procedures have been suggested. Mixture models such as the spike model is particularly recommended when the WTP distribution is asymmetric and a substantial fraction of

respondents choose not to consume the good [33], which is the case of 48.5% of respondents in our sample; while non-parametric models do not require any assumption on the WTP distribution at all.

**The spike model in the DBDC-CV framework.** In order to introduce the spike model, note that the "no-no" answer path in the DBDC-CV model (Eq 4) includes two types of answers: true zero WTP and false zero WTP, i.e. respondents that say "no" when the bid $t_i^l$ was asked for, but still have a positive WTP. Our third follow-up, maximum WTP question (see below) asks whether the individual would want to contribute at all to the investment "Scenario A" and, in that case, at what bid. This allows distinguishing between true and false zero WTP. Let $I_i^{NNY}, I_i^{NNN}$ be indicator functions such that in the group of "no-no" respondents, we have:

$$I_i^{NNY} = \boldsymbol{I} \left( ith \ respondent's \ answer \ who \ declared \ a \ maximum \ WTP \ such \ that \ 0 < max \ wtp_i < t_i^l \right) \tag{9}$$

$$I_i^{NNN} = \boldsymbol{I} \left( ith \ respondent's \ answer \ who \ declared \ to \ be \ willing \ to \ pay \ nothing \right)$$

Taking into account Eq 9, the extended log-likelihood function of the spike model is given by:

$$ln \, L(\theta) = \sum_{i=1}^{N} \{ I_i^{YY} \, ln[1 - G(t_i^u; \ \theta)] + I_i^{YN} \, ln[G(t_i^u; \ \theta) - G(t_i^0; \ \theta)] + I_i^{NY} \, ln[G(t_i^0; \ \theta) - G(t_i^l; \ \theta)]$$

$$+ I_i^{NNY} ln[G(t_i^l; \ \theta) - G(0; \ \theta)] + I_i^{NNN} ln[G(0; \ \theta)] \} \tag{10}$$

Assuming that the WTP is distributed as a logistic function on the positive axis [25] and, for the sake of clarity, that the model only includes the constant term and the bid, i.e. $\bar{X} = 1$ and $\theta = (\beta_1, \beta_{bid})$, it yields that:

$$G(t_i; \ \theta) = \begin{cases} [1 + \exp(\beta_1 - \beta_{bid} t_i)]^{-1} & if \ t_i > 0 \\ [1 + \exp(\beta_1)]^{-1} & if \ t_i = 0 \\ 0 & if \ t_i < 0 \end{cases} \tag{11}$$

where $\beta_1$ is the constant of the model and $\beta_{bid}$ is the coefficient on the bid. The spike, which is the empirical estimator of the probability of observing a zero WTP in the sample, is defined by $[1 + exp(\hat{\beta}_1)]^{-1}$; while the mean WTP in the spike model can be calculated as follows:

$$Mean \ (WTP) = \left( \frac{1}{\hat{\beta}_{bid}} \right) ln[1 + \exp(\hat{\beta}_1)] \quad \left( for \ \hat{\beta}_{bid} > 0 \right) \tag{12}$$

When covariates are added to the model, $\beta_1$ (the vector of coefficients associated with covariates) is replaced by $\bar{X}\hat{\beta}_1$ in Eqs 11 and 12, where, as before, $\bar{X}$ is the vector containing the sample means of covariates.

**The non-parametric maximum WTP.** As mentioned above, we asked respondents a third follow-up open-ended question about their maximum WTP: "*What is the maximum annual amount you would pay as a taxpayer for Scenario A?*" We required that the amount (in EUR) stated by the interviewee was coherent with the answers of the two previous DBDC-CV questions to avoid inconsistency in the sequence of replies [34]. We used this question to estimate the average maximum WTP, a distribution-free non-parametric estimator of the sample expected maximum WTP.

## Sampling and administration

A multi-stage stratified random sampling strategy was adopted for creating a representative sample of French citizens. In the first step, the sample was built so as to strictly mirror the population of French citizens in terms of gender, age, education, income, and region of residence (Table A in S5 File). According to SP guidelines [16], internet-based methods for surveys depend on issues of representation of the given population, and it is related to the level of computer literacy or the degree of connectivity across sociodemographic groups. We took this indication on board and in the second step the French population was split into people with internet access (88%) and those without internet access (12%) to customize the interview modality: CAWI (computer aided web interviews) for web users and CAPI (computer aided personal interviews), i.e. face-to-face, for non-web users (12%). The final sample was the sum of these two sub-samples and, in addition, it permits the separate analysis of the region of Auvergne-Rhône-Alpes (which hosts CERN) from the rest of the country.

The polling company conducted the survey in February 2018 yielding 1,005 valid interviews. Each version of the questionnaire was randomly allocated to a sub-sample of 201 people, whose internal demographic distribution mirrors that of the entire sample.

## Results

### Participants: Socioeconomic socioeconomic traits, personal interests and perceptions

In the sample 51% of respondents were female. 14.7% were aged less than 24; the remaining share were aged between 25 and 34 (17%), 35 and 44 (18%), 45 and 54 (19%), 55 and 64 (17.5%), and more than 65 (14%). Respondents with a high education level (i.e. at least a university degree) were more than 30%. The distribution of respondents according to the annual net income was 29.1% in the lower category of "less than EUR 15,000"; 29.3% earned between EUR 15,000 and 22,000; 17.3% between EUR 22,000 and 28,000" and 24.2% was in the highest category of "More than EUR 28,000". 12.5% of respondents lived in the region of Auvergne-Rhône-Alpes where part of CERN is located (Table A in S5 File). Employed people made up 60% of the sample, unemployed were 12.7%, and retired 18.1%. Students were 9.7% of the sample. Most people lived in an urban area or in the suburbs (60.2%) and belong to a family with 2 or 3 members (51.4%) (Table B in S5 File).

Section A of the questionnaire asks about general interests and opinions about the importance of scientific research in general, i.e. not necessarily related to CERN. This set of questions was asked before the exposure to the information material about CERN. Figure A in S5 File shows that most respondents declared that they were interested in arts and culture (57%), followed by environment (55%), and sports (44%) (multiple answers were possible). Scientific disciplines occupy the last positions with 37% declaring that they were interested in medicine, 23% in physics, and 20% in geology. According to French people, "improving health and quality of life" is the primary reason of importance for carrying out scientific research (93%), followed by "ensuring the future of next generations" (87%), and "increasing the human knowledge about the nature and origin of the universe" (81%). 65% of the sample agrees that "scientific research is important for supporting economic growth and employment" (Figure B in S5 File).

CERN was known, before submitting the information material, by 46% (466) of respondents (Fig 3), among whom 72% (321) had a positive perception of CERN, 26% neutral, and 2% declared that they had a negative perception.

Section B focussed on CERN and its research activity. The majority of questions in this section were asked after all respondents had looked at the information material (Figure C in S5

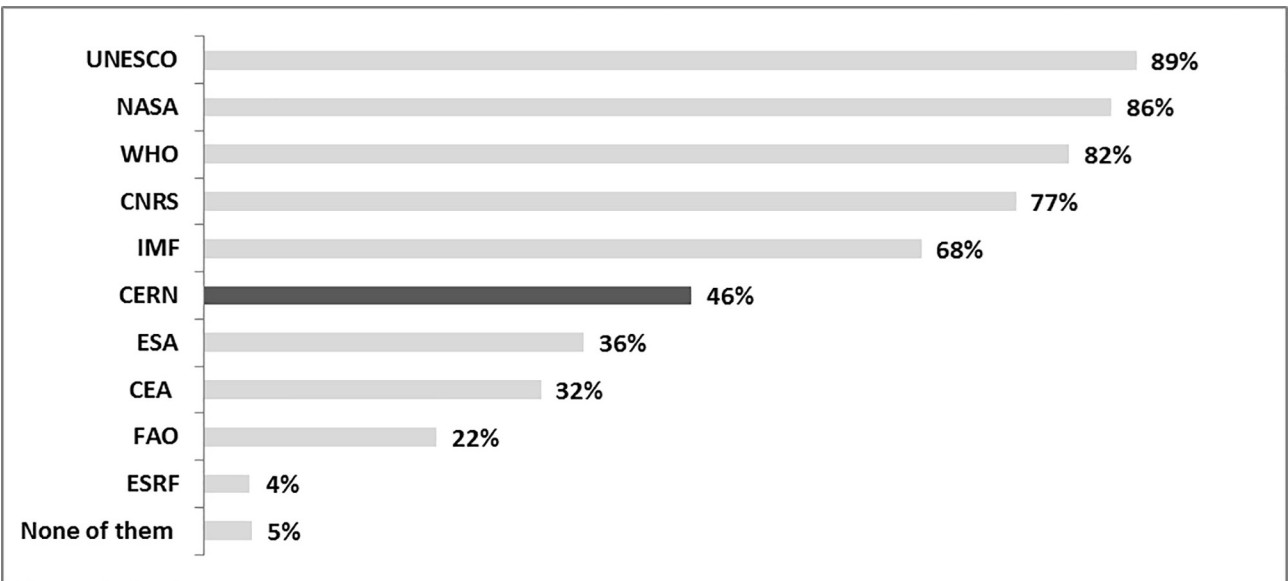

**Fig 3. Knowledge of CERN (*N* = 1,005).** Question A.4. Which one of the following organizations have you heard about? UNESCO (United Nations Educational, Scientific and Cultural Organization); NASA (National Aeronautics and Space Administration); WHO (World Health Organization); CNRS (French National Center for Scientific Research); IMF (International Monetary Fund); CERN (European Organization for Nuclear Research); ESA (European Space Agency); CEA (French Alternative Energies and Atomic Energy Commission); FAO (Food and Agriculture Organization); ESRF (European Synchrotron Radiation Facility). English translation.

File). More than 80% of them agreed that research at CERN advances "knowledge of the universe", "develops products for improving quality of life" (79%), and "helps develop technologies for medical applications" (77%). In addition, 68% thought that "research at CERN should be intensified in the coming decades".

Respondents were also asked to write down what aspects of CERN they appreciate, or they do not (Table C in S5 File). In line with the previous evidence, words such as "research lab", "scientific research", "big-bang", "universe", "particle physics", "particle accelerators" were cited by most interviewees (516 citations) followed by "medical research" (180), and "development and innovation" (145). Amongst aspects that people do not like there is "nothing" (576 citations), the fact that "CERN is a too difficult, complex subject" (68), or it is "dangerous" (67). Figure D in S5 File additionally shows that 83% of respondents agreed that "scientific research at CERN is important for everybody in the world"; in contrast, 29% thought that "it is important for scientists only" or "for the region where CERN is based" (26%). As a final point (after the exposure to the information material), the whole sample of interviewees were again asked to express a general perception of CERN. Out of 1,005 respondents, 78% (784) had a positive perception, 19% said they were neutral and 3% declared that they had a negative perception. The distribution of such answers is statistically different from the previous one (Pearson's $\chi^2$ = 301.4; *p-value < 0.01*) suggesting that the exposure to the two-page document about CERN and the movie reduced the percentage of neutral perceptions in favour of either positive perceptions (+6%) and negative ones (+1%)

## Estimation of the WTP for particle physics research at CERN

In accordance with the DBDC-CV elicitation format, Fig 4 presents the distributions of the responses for each triple combination of bids (initial, lower, and higher one) ranging from EUR 3 to EUR 100. Percentages are reported in circles. As expected, the higher the bid asked

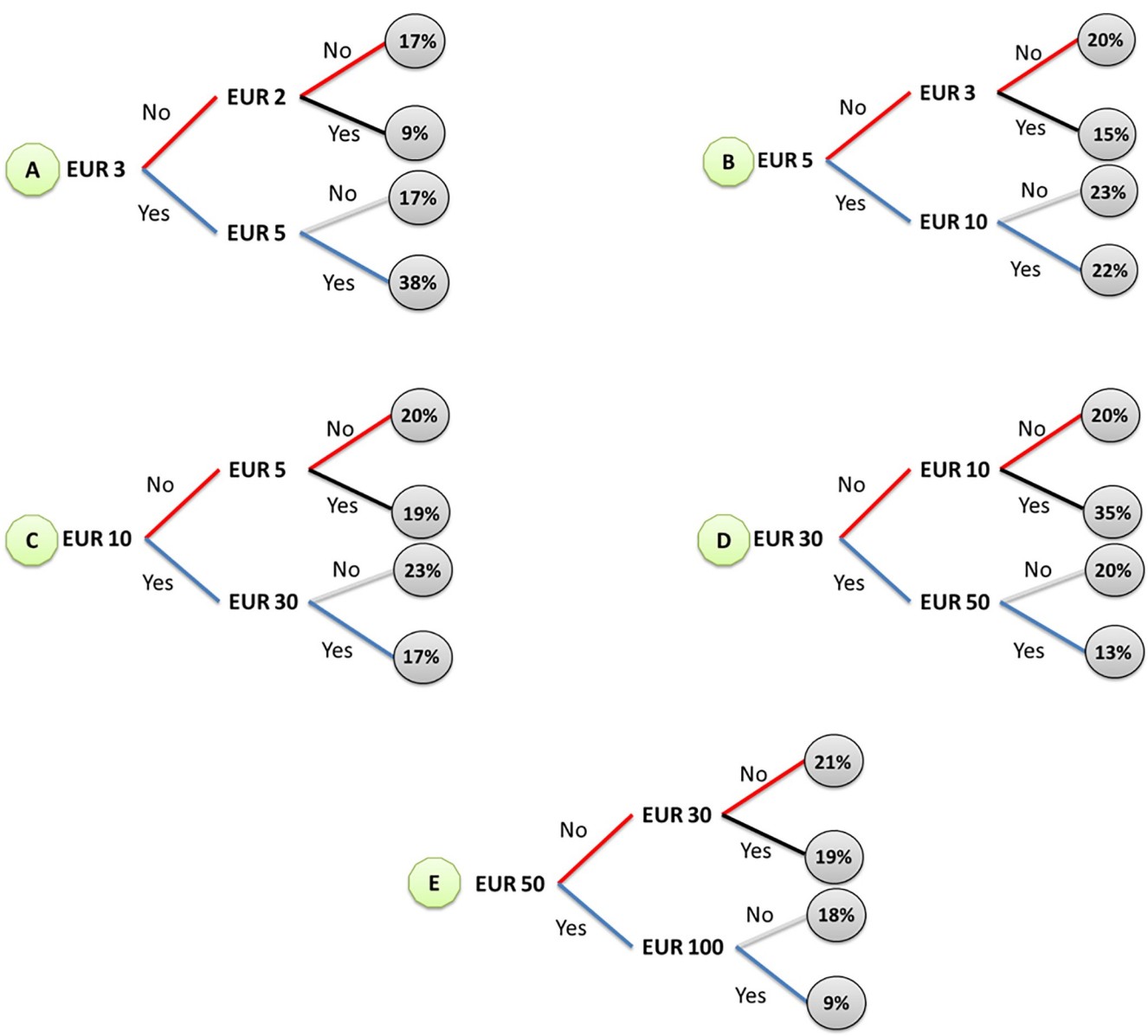

**Fig 4. Percentage distribution of the WTP answers by triple combination of bids.** The distribution is net of protest answers and outliers (*N* = 965). Letters A, B, C, D, and E in light green denote the version of the questionnaire; answers paths are: "No-no" (red); "no-yes" (red-black); "yes-no" (blue-light grey), "yes-yes" (blue). Grey circles display percentages.

the lower the probability of a "yes-yes" answer. The percentage of "yes-yes" decreases from 38% when the initial bid is EUR 3 to 9% when the initial bid is EUR 50. In contrast, the percentage of "no-no" increases as the bid asked increases.

The motivation why people were willing to pay, or why they were not, was asked and a number of protest bids were recorded (Table D in S5 File). Among people with zero WTP, people who are "against government-funded programs" and those who are "against international organizations" were identified as protestors. There were 29 and they were excluded from the calculation of the mean WTP. These reasons indicate protest answers and are not valid for justifying a zero WTP [35]. In addition, for conservative reasons and among people who stated a positive WTP, we excluded 12 outliers, i.e. those people with a ratio between the

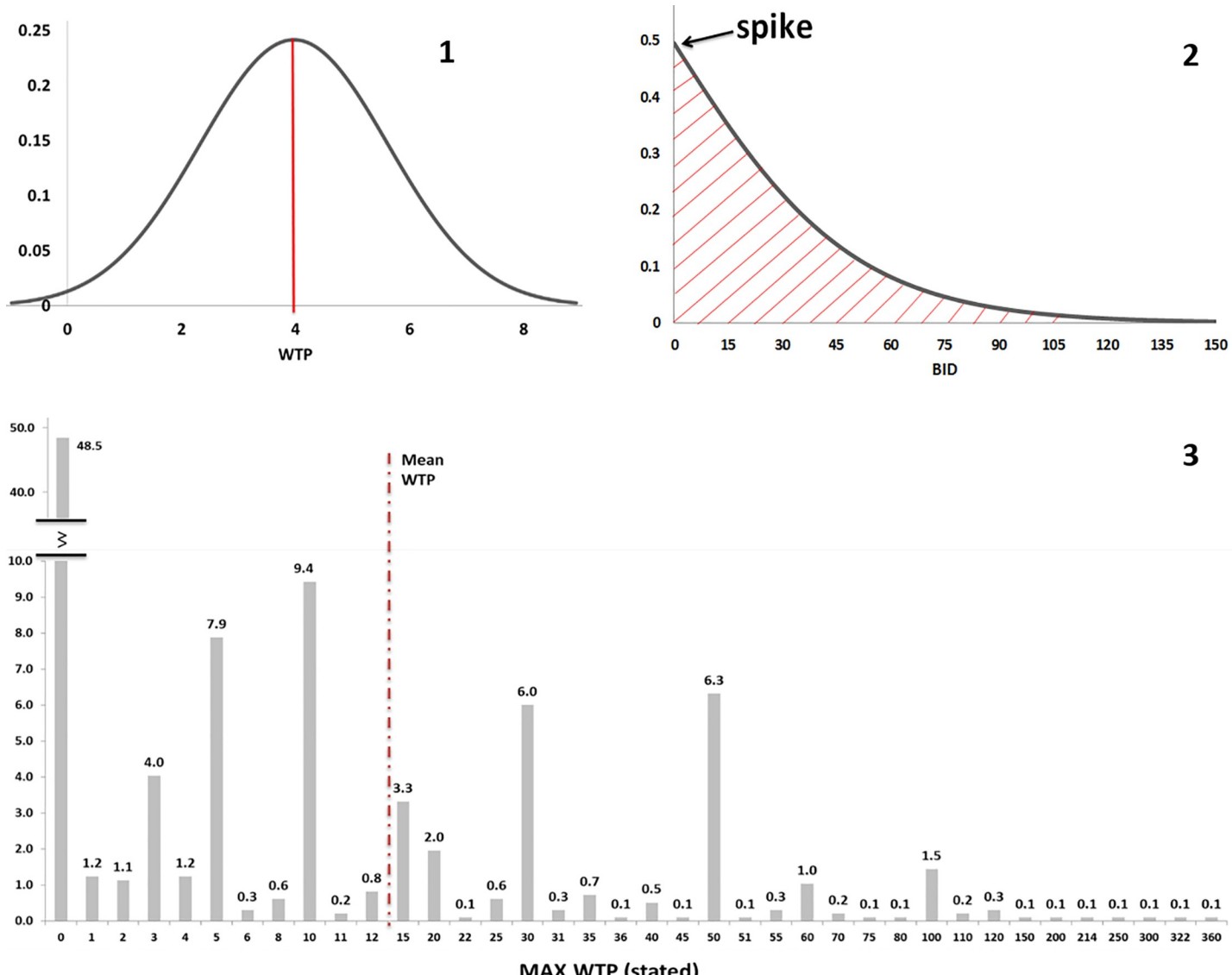

**Fig 5. Mean WTP (red) from Model 1, 2, and 3.** The WTP empirical probability distribution function from DBDC-CV model (**1**); The WTP empirical survival function according to the spike model (**2**), and the distribution (%) of the maximum WTP as stated by respondents (**3**). The mean WTP from model 3 includes zero answers.

stated maximum WTP and their income higher than 0.5%. As a result of these cuts at lower and upper tails of the WTP distribution, we ended up with a trimmed sample of 965 valid WTP answers. Fig 5 reports the estimation of the mean WTP from the DBDC-CV model, the spike model, and the non-parametric maximum WTP. In each model, only the bid is used as covariate and results are net of protest answers and outliers. Since dropping observations may lead to selection bias [36, 37], we tested the impact of our cuts on the mean WTP by performing supplementary analyses reported in Table A in S6 File. These cuts do not substantially influence the estimation of the WTP. For the sake of transparency, the Table B in S6 File also contains an estimation of the mean WTP according to the SBDC-CV procedure, which is recommended by the NOAA panel because of its incentive compatibility and lower exposure to anchoring.

**Table 1. Mean WTP (N = 965).**

|  | Model 1 (Double bounded dichotomous choice—DBDC) | Model 2 (Spike) | Model 3 (Sample mean) |
|---|---|---|---|
| **Mean WTP** | 3.97** | 16.93*** | 13.47*** |
| (Std. err.) | 1.65 | 1.17 | 0.93 |
| CI[a] | [0.74; 7.2] | [14.63; 19.22] | [11.64; 15.30][b] |
| Observations | 965 | 965 | 965 |

Notes

*p<0.1

** p<0.05

*** p<0.01.

[a]Confidence intervals.

[b]Bootstrapped confidence intervals with 1,000 replications.

Model 1 presents results from a conventional DBDC-CV model estimated from responses to the first two bids, assuming standard normal distribution [38] and applying Eq 8. It yields an estimated mean WTP of EUR 3.97 per person per year, which is statistically different from zero with *standard error = 1.65 and p-value = 0.016* (see Fig 5, panel 1, Table 1 and Table A in S7 File, Column 1). Model 2 resorts to a parametric spike model in order to fit the high percentage of zero willingness to pay [25] stated by French citizens. Zero responses in DBDC-CV can be interpreted as a corner solution of the consumers' utility-maximization when the good to be valued does not contribute at all to the individual's utility. In our sample, this was the case for 48.5% of respondents. The spike model separates the share of "no-no" answers from the DBDC-CV question format in two groups: real-zeros and a non-zero positive WTP. The resulting WTP distribution has two parts: a spike at zero, which represents the share of real-zeros and a truncated distribution for non-zero positive bidders. The spike model recognizes the distributional break at zero and better fits our empirical WTP distribution. To implement the spike model, we exploited the third follow-up maximum WTP question that tells us whether the respondent has a positive demand for CERN investment (zero otherwise). Assuming a logistic distribution and applying the Eq 12, the spike model returns a mean WTP of EUR 16.93, statistically different from zero at 1% level (*standard error = 1.17 and p-value <* 0.01) (see Fig 5, panel 2, Table 1 and Table B in S7 File, Column 1).

Model 3 estimates a non-parametric annual mean WTP from the maximum amounts stated by respondents in the third follow-up WTP question. The individual maximum WTP varies from EUR 0 to EUR 360, with a standard deviation of 29.80 (*standard error = 0.93*). Out of 965 respondents, 48.5% declared having a zero WTP (as mentioned above), while 51.5% were willing to contribute a positive amount. The non-parametric sample mean of the maximum WTP, including zero responses, is EUR 13.47 per person per year (Fig 5, panel 3).

## Drivers of the WTP

We turn then to examine how socioeconomic characteristics, awareness, and attitudes of respondents affect the respondents' WTP. Estimating the models with covariates is also common to test the internal consistency and theoretical validity in CV studies, including scope tests [39, 40]. Initially suggested by [15], "conventional" or "economic" (incremental) scope tests are based on the idea that consumers' utility should be sensitive to changes in the size and the scope of public goods: for example, more forests should be preferred to fewer forests, and in our context a higher probability of making discoveries is likely to be preferred to a lower

**Table 2. Econometric analysis: list of variables (N = 965).**

| Variable | Mean | Minimum | Maximum |
|---|---|---|---|
| Income | 2.36 | 1 | 4 |
| Male | 0.49 | 0 | 1 |
| Age (<35) | 0.32 | 0 | 1 |
| Education | | | |
| *high* | 0.33 | 0 | 1 |
| *medium* | 0.37 | 0 | 1 |
| Occupation status | | | |
| *student* | 0.10 | 0 | 1 |
| *employed* | 0.59 | 0 | 1 |
| *retired* | 0.18 | 0 | 1 |
| Family size (> 3 members) | 0.27 | 0 | 1 |
| Rhône-Alpes | 0.11 | 0 | 1 |
| Awareness of CERN | 0.46 | 0 | 1 |
| Scientific interest | 1.34 | 0 | 5 |
| CERN permits an increase in knowledge of the universe | 0.82 | 0 | 1 |
| The research activity at CERN should increase in the coming decades | 0.68 | 0 | 1 |

probability, and so on. Accordingly, higher WTP should be expected when people are offered more than less. However, passing the economic scope test is not a sufficient condition for the validity of a CV survey [39]. Behavioural intentions, including WTP, are influenced by a person's socioeconomic traits, attitudinal and cognitive dimensions, as well as by individual beliefs and interest concerning public good. Regardless of whether contingent values show economic scope or not, respondents with more interest, knowledge, and attitude towards a commodity are likely to pay more for that commodity. Therefore, exploring the impact of these WTP conditioning factors is fundamental in showing the validity of the CV experiment and its compatibility with economic and social psychological theory. Following earlier literature on people's attitude and interest in science, technology, and research [9, 41], as well as psychological theory [39], we focussed on three set of covariates: financial factors (income), demographic characteristics (gender, age, level of education, occupation status, family size, region of residence), and personal interest and values (awareness of CERN, having interest in scientific subjects, and cognitions, i.e. personal beliefs about CERN research).

We pre-treated the survey responses to obtain the variables that were entered in the econometric analysis and included these variables in both the DBDC-CV model (Table A in S7 File, Columns 2–7) and in the spike model (Table B in S7 File, Columns 2–8). Moreover, we tested correlation between each variable and the stated maximum WTP by resorting to Spearman's rho correlation coefficient (Table C in S7 File). The variables are (see Table 2 for their statistical distribution):

- *Income* is coded as 1 if income is lower than EUR 15,000, 2 if it is between EUR 15,000 and 22,000; 3 if it is between EUR 22,000 and 28,000; 4 if it is more than EUR 28,000.

- *Male* is a binary variable, taking the value of 1 if the respondent was male and 0 otherwise;

- *Age (<35)* is a binary variable taking the value of 1 if the respondent was aged less than 35 and 0 otherwise;

- *Education—high* is a binary variable, taking the value of 1 if the respondent had a high level of education and 0 otherwise;

- *Education–medium* is a binary variable, taking the value of 1 if the respondent had a medium level of education and 0 otherwise;

- *Occupation status–student* is a binary variable, taking the value of 1 if the respondent was a student and 0 otherwise;

- *Occupation status–employed* is a binary variable, taking the value of 1 if the respondent was employed and 0 otherwise;

- *Occupation status–retired* is a binary variable, taking the value of 1 if the respondent was retired and 0 otherwise;

- *Family size (> 3 members)* is a binary variable, taking the value of 1 if the respondent belonged to a family with more than 3 members and 0 otherwise;

- *Rhône-Alpes* is a binary variable, taking the value of 1 if the respondent lived in Auvergne-Rhône-Alpes region and 0 otherwise;

- *Awareness of CERN* is a binary variable, taking the value of 1 if the respondent already knew CERN before the exposure to the information material and 0 otherwise;

- *Scientific interest* is a composite categorical variable. It was built on the answers shown in Figure B in S5 File and specifically being interested in scientific subjects such as medicine, biology, astronomy, physics, and geology. Originally submitted as a 5-point Likert scale, for the purposes of data analysis, each answer was transformed into a binary variable by assigning the value of 1 if the respondent was "enough or "very" interested in that subject and 0 otherwise. Then, the "scientific interest" variable was obtained as the sum of the five binary items (each one reflecting the interest in one of the scientific subjects) and its value ranges from 0 to 5. The value of 0 means that the respondent is not interested in any of the subjects; in contrast, the value of 5 reflects the interest in all the subjects. Thus, the higher the value of this variable, the higher the number of scientific subjects the respondent is interested in;

- *CERN permits an increase in knowledge of the universe* is a binary variable, taking the value of 1 if the respondent agreed or strongly agreed with that statement and 0 otherwise;

- *The research activity at CERN should increase in the coming decades* is a binary variable, taking the value of 1 if the respondent agreed or strongly agreed with that statement and 0 otherwise.

Table A in S7 File reports the whole set of estimates from the double bounded dichotomous choice model, while Fig 6 below plots coefficients from specification 4 (which includes all the respondents' socioeconomic traits) and specification 7, which adds personal interests and beliefs. It was estimated by using the Stata command "doubleb" [42] which assumes standard normal c.d.f. in Eq 5. The guidance for SP studies [16] suggests that the WTP analysis should primarily assesses only the influence of the bid variable, apart from effects of other covariates; accordingly, seven specifications are presented. In the first one (Column 1), we only include the constant and the bid. The latter is not displayed in the table because the command "doubleb" uses bids to create bounds to the WTP [42]. As mentioned above, the mean WTP in EUR per person per year amounts to EUR 3.97. In the second specification (Column 2), we only included the income variable to test its influence on the WTP, which is positive (*p-value < 0.01*). The third specification (Column 3) shows that a high level of education is significantly and positively associated with the WTP for research activities [10] (*p-value < 0.05*); in contrast, the gender, age and the place where people live (i.e. living in the neighbourhood of CERN in Auvergne-Rhône-Alpes with respect to live away from CERN) do not influence the

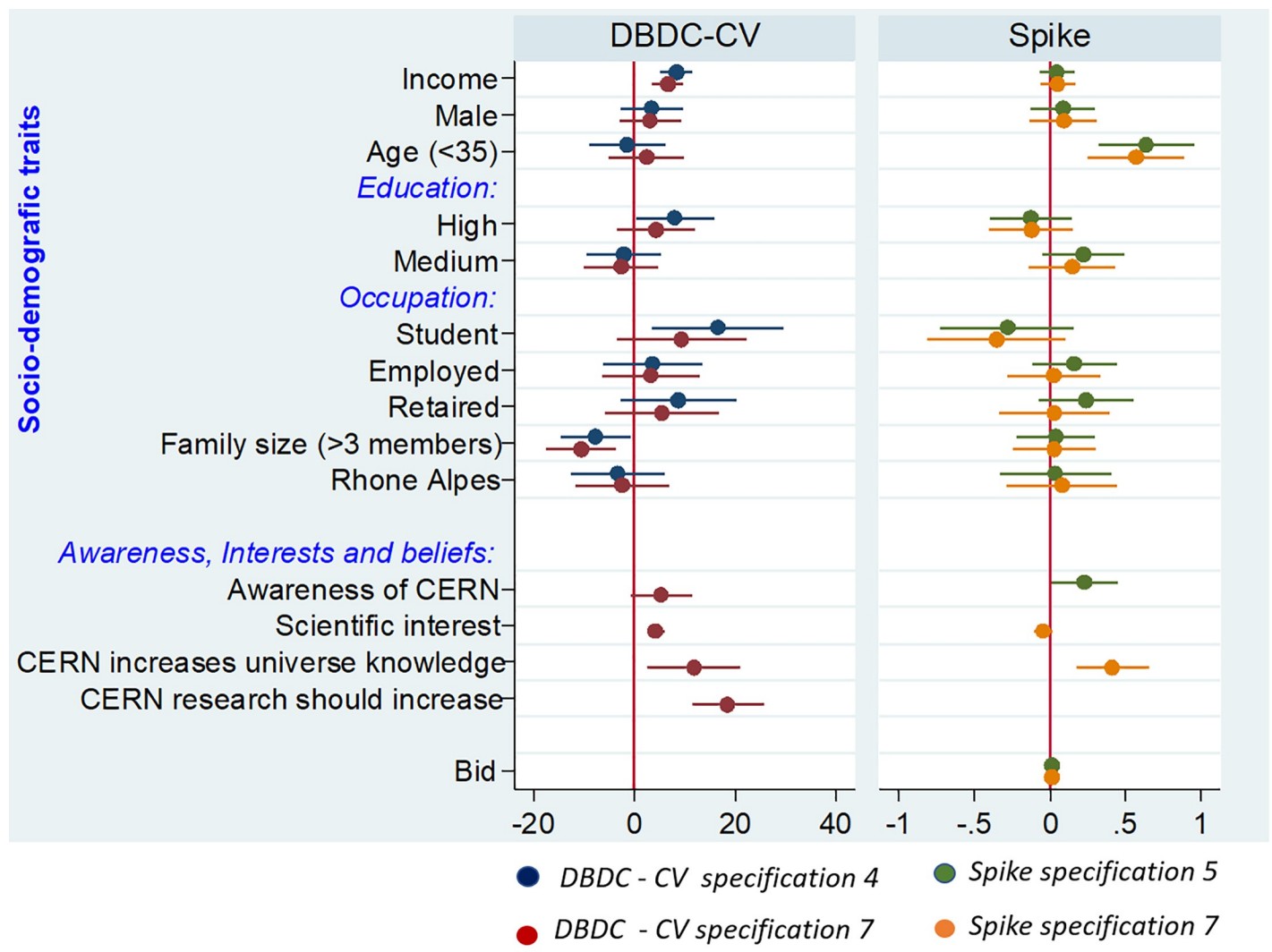

**Fig 6. Drivers of WTP.** Plot of the coefficients from the double-bounded dichotomous choice (DBDC-CV) specification 4 (dark blue), and specification 7 (red) and from the spike specification 5 (green) and specification 7 (orange). The STATA command "coefplot" was used to obtain the graphs [43].

respondents' willingness to contribute for basic research. One may argue that the occupational status and the family size may affect somehow the WTP. We tested these hypotheses in the fourth specification (Column 4). Students are willing to pay more than other people, while being a member of a large family negatively influences the WTP: as family size increases, budgets tighten and WTP is likely to fall. This may be the case of projects, such as investments in basic research, which do not bring immediate or easily perceptible social welfare improvements to people. Awareness of CERN (Column 5) and having some interest in some scientific subjects (Column 6) positively and significantly affect the WTP (*p-value < 0.01*). Column 7 presents the full model, which variables capturing thoughts such as "CERN permits an advance in the knowledge of the universe" or "research should increase in the coming time" are jointly plugged into the same model. We found statistical evidence on the contribution of these variables on WTP for investing in particle physics research projects. The likelihood ratio test in each specification indicates that the estimated equation is statistically different from zero at the 1% level. Put differently, variations in the socioeconomic characteristics and cultural traits of

respondents explain a good proportion of variability in the WTP. This is somehow corrobo-rated by the Akaike information criteria (AIC) and the Bayesian information criteria (BIC): the specification in Table A in S7 File, Column 7 returns the minimum AIC and BIC values.

Results from the spike model are visualised in Fig 6 and fully reported in Table B in S7 File, which follows the same logic as Table A in S7 File. All estimations were implemented in Stata with the command "spike", which assumes a logistic c.d.f. in Eq 10. Although comparisons between the two models are not straightforward because of different assumptions, the spike model confirms the results of the DBDC-CV model, with the exception of three findings. Firstly, the mean WTP is positive but much higher than in the DBDC-CV model: it amounts to EUR 16.93 in the first specification (Table B in S7 File, Column 1). Such a result is in line, for instance, with [25]. Secondly, "being a student" loses its predictive power in explaining var-iations in WTP in favour of "being aged less than 35" (Fig 6). The high and statistically signifi-cant correlation between those variables may explain this result (*Spearman's rho correlation coefficient = 0.47; p-value <0.01*). Thirdly, the income variable has no effects on WTP "yes" when the variables capturing awareness of CERN and respondents' beliefs on research activity at CERN are included in the model (Table B in S7 File, Columns 5, 7, and 8). Note that in the spike model the specification (Column 8) was added because there was no convergence of the algorithm when the variables "CERN permits an advance in the knowledge of the universe" or "research should increase in the coming time" were jointly plugged into the same model.

As far as the non-parametric estimation of the WTP, the sample mean of the stated maxi-mum WTP, including zero answers, yields the figure of EUR 13.5 per person per year. In line with the DBDC-CV and the spike models, a bivariate correlation between the maximum WTP as declared by respondents and each independent variable reported in the econometric analysis was carried out. Correlations, tested with Spearman's rho correlation coefficient are reported in Table C in S7 File. Results are consistent with those discussed above. Income, high-level educa-tion, interest, and a favourable attitude towards science are strongly associated with the WTP. Overall, these results indicate consistency with economic and social psychological thinking.

As stated above, we estimated the mean WTP from specifications that consider only the constant and the bid either using the DBDC-CV model or the spike one. The mean WTP goes to EUR 4.02 (*p-value* < 0.01) when the income is added as covariate in the DBDC-CV model and increase to about EUR 37.06 (*p-value* < 0.01) in the spike model.

The mean WTP varies even more when additional covariates are jointly plugged into the same specification. It turns out to be slightly negative and not statistically different form zero in the DBDC-CV model (Table A in S7 File, Columns 4–7) and very high (more than EUR 40, statistically different from zero at 1% level) in the spike model (Table B in S7 File, Columns 3–8). While extended models with covariates are useful tools to test the determinants of WTP, simple specifications including only the bid and income remain our preferred ones for infer-ring the mean WTP for particle physics research at CERN. In the CV literature such specifica-tions are most widely used to elicit the WTP because they essentially validate the basic axioms of preferences, including the non-increasing relationship between the proportion of people who would pay, the amount of bid asked and the relevance of budget constraints [16]. Actually, as we have a pure public good, whether considering income elasticities to infer the WTP remain an issue since for public goods we do not necessarily expect WTP to have as large as income elasticity as we would with respect to demand for a private good [44].

## Discussion and conclusion

This paper addressed the question of whether the public is willing to pay to support fundamen-tal science investments, whose discoveries, generally, do not have any immediate application

or utility for the average taxpayer. The latter, however, supports the costs of Big Science investments. In this analysis, hence, we sought to explore whether a non-use value of science as a public good can be empirically estimated.

Our research produces three main findings. First, we show, for the first time, that CV experiments may be applied to study the public good value of scientific research, similarly to environmental and cultural goods. This potentially opens up a new avenue in science policy empirical analysis. Indeed, while asking the layman to opine on technical issues may raise scepticism, in open democratic societies this is not unheard of. For instance, in numerous countries there have been referendums on whether or not to shut down nuclear power stations, which is a quite technical issue.

Second, the analysis suggests that with any of the estimated model citizens' WTP is in excess of what they implicitly pay to CERN as taxes: in 2017 the per capita implicit tax contribution of French adults (18+) to CERN was EUR 2.7 per year against an estimated WTP range of 3.97–16.93. While these WTP results hold for French citizens, it would be unwise to speculate on their external validity in other CERN Member States' because cultural values and socioeconomic conditions may be different.

Third, we examine the drivers of the WTP for particle physics research. WTP is positively and significantly correlated with income, which is not necessarily expected for a pure public good. A high level of education, i.e. having at least a university degree, is a strong driver of the WTP for research activities at CERN. The positive and statistically significant coefficients on being a student in the DBDC-CV model and on being aged less than 35 years in the spike model suggests that, in general, being young increases the probability of financially supporting CERN as compared to older people. The analysis also indicates, *ceteris paribus*, that the preferences of French people in supporting investments at CERN are similar regardless of the place they live (hence regardless of perceived local benefits related to tourism, employment, and procurement). The variables that capture personal values and interests towards science are, as expected, strong drivers of being willing to pay. In particular, the high correlation of the citizens' WTP with previous awareness of CERN suggests that outreach activities of basic research significantly contribute to the public good value of science. In a precise sense: outreach of science creates social value [45].

## Supporting information

**S1 File. Pilot experiments and pre-tests.**
(PDF)

**S2 File. The questionnaire (in French and in English).**
(PDF)

**S3 File. Information material: Two-page description of CERN (in French and in English).**
(PDF)

**S4 File. The informed consent.**
(PDF)

**S5 File. Respondents' socioeconomic traits, personal interests and perceptions of CERN.**
(PDF)

**S6 File. Supplementary analyses.**
(PDF)

**S7 File. Drivers of the WTP.**
(PDF)

**S8 File. Anonymized survey data.**
(XLS)

## Acknowledgments

We are grateful for their comments to: Glenn Blomquist (Gatton College of Business and Economics—University of Kentucky), Gelsomina Catalano (CSIL and University of Milan), Stefano Forte (University of Milan and Italian National Institute for Nuclear Physics), Alejandro Lopez-Feldman (Centro de Investigacion y Docencia Economicas, CIDE), Per-Olov Johansson (Stockholm School of Economics), Bengt Kriström (Swedish University of Agricultural Sciences), Giovanni Perucca (Politecnico di Milano), Riccardo Scarpa (Durham University), and David L. Weimer (Robert M. La Follette School of Public Affairs University of Wisconsin–Madison). We are also grateful to several experts at CERN and particularly to Michael Benedikt, Panagiotis Charitos, Johannes Gutleber, and Olivier Martin. Comments by attendees at our sessions at the following conferences are gratefully acknowledged: Annual Conference of the Society for Benefit-Cost Analysis in March 2019, George Washington University, Washington DC, USA; FCC Host State Realization working group meeting in June 2018, CERN, Switzerland; 104th Annual Congress of the Italian Society of Physics in September 2018, University of Calabria, Italy; 57th Annual Conference of the Italian Economic Association (SIE) in October 2016, Bocconi University, Italy. We are also thankful to all the interviewees for their participation in the survey and pilot experiments.

## Author Contributions

**Conceptualization:** Massimo Florio, Francesco Giffoni.

**Data curation:** Francesco Giffoni.

**Formal analysis:** Francesco Giffoni.

**Funding acquisition:** Massimo Florio.

**Investigation:** Francesco Giffoni.

**Methodology:** Francesco Giffoni.

**Project administration:** Massimo Florio.

**Software:** Francesco Giffoni.

**Supervision:** Massimo Florio.

**Validation:** Francesco Giffoni.

**Writing – original draft:** Massimo Florio, Francesco Giffoni.

**Writing – review & editing:** Massimo Florio, Francesco Giffoni.

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
