## [Decision Letter · Decision Letter 0]

27 Nov 2019

PONE-D-19-28867

A contingent valuation experiment about future particle accelerators at CERN

PLOS ONE

Dear Professor FLORIO,

Thank you for submitting your manuscript to PLOS ONE. Let me first warn you that your original editor was unable to complete handling your paper and I stand in. After having read the paper, I had the impression it had an important contribution, even if a bit too deeply buried in the paper structure. This sentiment is shared by the Referees (whose two reports you will find below). Hence, I would like to invite you to revise the paper and resubmit it for PLOS ONE.

In addition to the comments of the Referees, please, revise also the structure of the paper. I admire your drive for transparency and PLOS ONE is definitely not the journal to discourage even lengthy, but important appendices. However, I would advise that the paper starts from the puzzle, states the intended contribution, explains how this contribution is achieved and then discusses the findings. This should be a self-contained unit, from start to end. The rest of your supplementary materials may be relegated to the appendices, but it should be clearly indicated in the main text that a given appendix provides additional insights. At this point, the introduction is too brief (especially for a general audience), and the rest of the paper has an illogical structure (first some results, then some methods, then more results -- confusing. In particular, I am not sure if pre-tests have to be described in such a detail, but I leave it to your discretion, once you revise the main narrative of the paper.

Also, please, follow PLOS ONE guidelines on paper formatting (referencing in particular)

We would appreciate receiving your revised manuscript by Jan 11 2020 11:59PM. To enhance the reproducibility of your results, we recommend that if applicable you deposit your laboratory protocols in protocols.io, where a protocol can be assigned its own identifier (DOI) such that it can be cited independently in the future. For instructions see: http://journals.plos.org/plosone/s/submission-guidelines#loc-laboratory-protocols

We look forward to receiving your revised manuscript.

Kind regards,

Joanna Tyrowicz

Academic Editor

PLOS ONE

Journal Requirements:

2. Please include a copy of the questionnaire in the original language as supporting information, in addition to the English version you already provided.

Reviewers' comments:

Reviewer's Responses to Questions

**Comments to the Author**

1. Is the manuscript technically sound, and do the data support the conclusions?

Reviewer #1: Yes

Reviewer #2: Yes

2. Has the statistical analysis been performed appropriately and rigorously? 

Reviewer #1: Yes

Reviewer #2: Yes

3. Have the authors made all data underlying the findings in their manuscript fully available?

Reviewer #1: Yes

Reviewer #2: Yes

4. Is the manuscript presented in an intelligible fashion and written in standard English?

Reviewer #1: Yes

Reviewer #2: Yes

5. Review Comments to the Author

Reviewer #1: General comments:

This article provides novel findings on the WTP for basic science in France. It uses an adequate method (CV), carefully administered and pre-tested. The paper is well and clearly written. However, the balance between the main text and the supporting information should be reassessed. In particular, the methodology, descriptive statistics and empirical analysis deserve entire chapters in the main text.

Major comments:

The treatment of protest should be done more carfully, see my comment below.

Methodology and empirical analysis, already present in the supporting information, should be integrated in the main text.

Specific comments:

line 115: dropping observations may lead to selection biases. One should be more cautious here. Are protestors and outliers significantly different from "normal" bidders? Do these special characteristics have an impact on WTP? See Garcia et al. (2009) or Borzykowski et al. (2017)

lines 124 to 128: This formulation is too close from Borzykowski et al. (2018), please modify or quote.

Line 158: Why do you talk about scope test here? Did you perform any? Do you analyse internal consistency later? I know that you do, but it is not clearly stated in the text, so please explain exactly what you consider to be an internal consistency test and later in the text, please specify if you find consistency or not.

line 165: What do you mean by "tested bivariate relationships"? Did you perform a linear regression between 2 variables or just correlation? Please explain

line 167: Why is the relationship between income and WTP for a public good inexpected? Please explain

line 178: Is awareness of CERN linked with the place respondents live? I assume that, the closer they live, the more aware they. This would lead to an issue of colinearity, which could explain the non-significant impact of distance. Don't you have a way to be more specific regarding the place of residence? In particular for face-to-face interviews, I assume that you have addresses.

line 192: I am surprised by the fact that the DBDC gives much smaller WTP. Any hypothesis why? What distributional assumption would you consider better?

line 249 and following: Was you sample representative of the student population?

line 273: Why and how were respondent likely to react differently with lump-sum vs. annual payments?

line 283: "Would you BE willing to pay EUR" the "be" is missing

line 301: What do you mean by "binomial logit model with numerical integration"? What is the numerical integration about? line 479: How do you get a 100% response rate? This is misleading or doubtful

line 483-486: Not clear. Please rephrase

line 603: Please cite Borzykowski et al. (2018) here.

line 650: This estimator is affected by the anchor of the bids. How do you treat this issue? If you don't, you need to.

line 805: What robustness test are you talking about? Statistically, you cannot reject the alternative hypothesis. You do not reject the null hypothesis of mean equality.

Tables S6, s7: What is the preferred model? Could you provide the AIC / BIC and discuss it in the text?

Table S7: How do you explain the positive impact of the bid variable? What does this mean and how do you get positive WTP afterwards?

Table s8: How were the CI computed?

References:

Borzykowski, N., Baranzini, A., & Maradan, D. (2017). Y a-t-il assez de réserves forestières en Suisse? Une évaluation contingente. Économie rurale. Agricultures, alimentations, territoires, (359), 51-79.

Borzykowski, N., Baranzini, A., & Maradan, D. (2018). Scope effects in contingent valuation: does the assumed statistical distribution of WTP matter?. Ecological Economics, 144, 319-329.

Garcia S., Harou P., Montagné C., Stenger A. (2009). Models for sample selection bias in contingent valuation: Application to forest biodiversity. Journal of Forest Economics, vol. 15, n° 1, pp. 59-78.

DOI : 10.1016/j.jfe.2008.03.008

Meyerhoff J., Mørkbak M.R., Olsen S.B. (2014). A meta-study investigating the sources of protest behaviour in stated preference surveys. Environmental and Resource Economics, vol. 58, n° 1, pp. 35-57.

DOI : 10.1007/s10640-013-9688-1

Strazzera E., Genius M., Scarpa R., Hutchinson G. (2003). The effect of protest votes on the estimates of WTP for use values of recreational sites. Environmental and Resource Economics, vol. 25, n° 4, pp. 461-476.

Reviewer #2: The article is very interesting and it offers a novel approach to study attitudes to science and technology. It also includes a thorough description of the design, methodology and implementation procedure. Nevertheless, the statistical models are not clearly enough defined. It is possible that they are evident for readers familiarized with stated preferneces techniques, Contingent Valuation and the utility-theoretic framework, but not for researchers not familiarized with them. The intelligibility and interest of the article will improve with a brief and clear description of the models and of the utility-theoretic framework.

On the other hand, in Table S5, the authors provide the descriptive statistics of the variables of what they call the "Econometric analysis". It is quite strange and disinformative read that the variable "male" has a mean of 0.49 with a Standard Deviation of 0.50, that the mean of high education is 0.33 with a Standard Deviation of 0.47, etc. This may be a well stablished procedure in econometric analysis, but descriptive statitistic clearly stablish that the mean and standard deviation, as measures of central tendency, are only appropriate to describe cuantitative variables (continuous and discret) but not qualitative ones (dichtomous and ordinals). Although not the best election, the mean and standard deviation only may be considered informative in describing scientific interest, as this variable is ordinal with five levels. Obviously, this comment is "anechdotic" and, hence, does not discredit the high quality of the article, but it is advisable that the descriptive statistic adjusts to the measuring level of the variable.

Finally, an other to some extent irrelevant for the final results of the article, but somewhat disconcerting issue, is the decision to measure attitudes with a five points scale with a midpoint defined as neutral, to finally collapse the five categories into two, combining the neutral point with the "negative" categories (disagree and strongly disagree). First, if it is properly a neutral category, it should be considered equidistant to the positive and negative poles of the scale and thus might be logically included in the positive section of the scale. Second, there arefounded doubts in the literature concerning the real meaning of the neutral midpoint in attitude questions, as it may represents a combination of actual neutral responses, covert "I don't know" answers, or a satisficing strategy: the tendency to provide a reasonable response, not the best one; that is to say, the respondents tendency to select the "easy" option.

6. PLOS authors have the option to publish the peer review history of their article (what does this mean?). If published, this will include your full peer review and any attached files.

Reviewer #1: No

Reviewer #2: No

---

## [Author Response · Author response to Decision Letter 0]

28 Jan 2020

See the attached file 'Response to Reviewers'

---

## [Editor Report · Decision Letter 1]

11 Feb 2020

PONE-D-19-28867R1

A contingent valuation experiment about future particle accelerators at CERN

PLOS ONE

Dear Professor FLORIO,

thank you for submitting your revision to PLOS ONE. In the interest of clarity and readability of your text, I suggest two following changes.

Change #1. In the final part of your empirical analysis, you report three separate tables, each with full set of specifications, to make only a few notes. I find this ok (no need to make more points!), but perhaps it would be easier for the readers to follow your line of reasoning if you report a plot of marginal effects from all three outcome measures in one graph, space in the graph split by variables of interest. I have in mind something like Figure 7 in this source (if you use STATA: https://www.stata.com/meeting/germany14/abstracts/materials/de14_jann.pdf) or predicting gas mileage plot (if you use R: https://cran.r-project.org/web/packages/dotwhisker/vignettes/dotwhisker-vignette.html). It can be vertical or horizontal, but my point is to shorten three tables to one graph that has all the content you require. You can (and should) relegate the full tables to the online appendix then.

Change #2. You use frequently the abbreviation DBDC-CV without ever (I think ever?) explaining it to the reader. Please, make the courtesy to the reader to make it clear below every table or figure what each abbreviation signifies, so that one did not have to browse through the whole text to find the meaning. 

Overall, I cannot ask you to make those changes other than through minor revision option, I hope you do understand that the changes I propose above are in the interest of making your paper more reader-friendly. You may consider your paper accepted, conditional on those minor changes.

We would appreciate receiving your revised manuscript by Mar 27 2020 11:59PM. To enhance the reproducibility of your results, we recommend that if applicable you deposit your laboratory protocols in protocols.io, where a protocol can be assigned its own identifier (DOI) such that it can be cited independently in the future. For instructions see: http://journals.plos.org/plosone/s/submission-guidelines#loc-laboratory-protocols

Please note while forming your response, that you may have the opportunity to make the peer review history publicly available. The record will include editor decision letters (with reviews) and your responses to reviewer comments. If eligible, we will contact you to opt in or out.

We look forward to receiving your revised manuscript.

Kind regards,

Joanna Tyrowicz

Academic Editor

PLOS ONE

---

## [Author Response · Author response to Decision Letter 1]

15 Feb 2020

See the attached file 'Response to Reviewers'

---

## [Editor Report · Decision Letter 2]

19 Feb 2020

A contingent valuation experiment about future particle accelerators at CERN

PONE-D-19-28867R2

Dear Dr. FLORIO,

We are pleased to inform you that your manuscript has been judged scientifically suitable for publication and will be formally accepted for publication once it complies with all outstanding technical requirements.

With kind regards,

Joanna Tyrowicz

Academic Editor

PLOS ONE
---

## [Editor Report · Acceptance letter]

2 Mar 2020

PONE-D-19-28867R2 

A contingent valuation experiment about future particle accelerators at CERN 

Dear Dr. Florio:

I am pleased to inform you that your manuscript has been deemed suitable for publication in PLOS ONE. Congratulations! Your manuscript is now with our production department. 

With kind regards,

on behalf of

Professor Joanna Tyrowicz 

Academic Editor

PLOS ONE